# Does A Velamen Radicum Effectively Protect Epiphyte Roots against Excessive Infrared Radiation?

**DOI:** 10.3390/plants12081695

**Published:** 2023-04-18

**Authors:** Calixto Rodríguez Quiel, Helena J. R. Einzmann, Gerhard Zotz

**Affiliations:** 1Functional Ecology Group, Institute of Biology and Environmental Sciences, University of Oldenburg, P.O. Box 2503, D-26111 Oldenburg, Germany; 2Smithsonian Tropical Research Institute, Panama City 0843-03092, Panama

**Keywords:** epiphytes, heat stress, light, roots, velamen radicum

## Abstract

Velamen radicum, a dead tissue at maturity, characterizes the roots of many epiphytes. Apart from a role in water and nutrient uptake, protection against excessive radiation in the upper forest canopy has also been suggested, but this function has never been critically assessed. To test this notion, we studied the roots of 18 species of Orchidaceae and Araceae. We defined thermal insulation traits of velamina by monitoring the temperature on the velamen surface and just below the velamen while exposing it to infrared radiation. We investigated velamen’s functionality-correlating morphological and thermal insulation traits. In addition, we investigated the viability of the living root tissue after heat exposure. The maximal surface temperatures ranged from 37–51 °C, while the temperature difference between the upper and lower velamen surface (i.e., ∆T_max_) ranged from 0.6–3.2 °C. We found a relationship of velamen thickness with ∆T_max_. Tissue viability was strongly affected by temperatures >42 °C, and no significant recovery after heat exposure was found. Thus, there is only limited support for an insulating function of velamen, but the data suggest considerable species-specific differences in heat tolerance. The latter could be a crucial determinant of the vertical distribution of epiphytes.

## 1. Introduction

Roots of terrestrial plants typically grow below ground, where they serve as the anchor of the above-ground shoot and are also responsible for the uptake and transport of water and nutrients [1]. The dynamics of temperature and moisture conditions in soil are usually rather slow [2]. In contrast, many epiphytic plants grow directly on the bark of their host tree, and thus face entirely different abiotic challenges. They can be exposed to intense radiation, and temperature and moisture can vary fast, frequently and drastically [3,4]. One conspicuous feature of the aerial roots of the species-rich taxa Orchidaceae and Araceae, which has attracted particular interest for a long time, is velamen radicum. It is a spongy, usually multiple epidermis that is dead at maturity, bordering externally on an exodermis with both impermeable, thick-walled cells and so-called passage cells [5]. Velamen is also found in numerous terrestrial taxa [6], but scientific attention has almost exclusively focused on velamen in epiphytic orchids. There is a wealth of information on anatomical and morphological characteristics of the velamen in this family [7,8,9], but the connection between form and function is still largely conjectural. In the case of other taxa with velamentous roots, such as many species of the genus *Anthurium* [6], data on anatomy and function are almost entirely missing.

There is a consensus that velamen serves multiple functions. In his thorough review, Pridgeon [5] proposed at least five functions, namely the uptake and storage of water and dissolved nutrients, reduction in transpiration, scatter of light for photoprotection, reduction in heat load and mechanical protection. Actual evidence is available for only a few of these functions. Relatively well studied is the role of velamen in nutrient uptake. For example, Zotz and Winkler [10] tested a notion originally put forward by Went [11]. As suggested, they found that charged ions were captured and temporarily immobilized in velamen and subsequently taken up into the living cortex. Later, Chomicki et al. [12] added UV protection to the list of functions. They provided experimental evidence that exposure to UV-B induces the synthesis of chalcone synthase in the roots, which leads to the accumulation of two UV-B-absorbing flavonoids, effectively protecting the photosynthetic root cortex.

Surprisingly, there is no information at all on the role of the velamen of aerial roots or on the amount of radiation reaching the underlying living cortex, which is frequently green and capable of photosynthesis [13,14]. There is also no documentation backed by any data that suggests the protective role of roots against heat load due to intense radiation, particularly in the upper crown of tropical trees.

The current study investigated the possible insulating effect of velamen by comparing the temperature increases in the velamen surface and the outermost layer of the living cortex when exposed to near-infrared radiation (NIR) of an intensity typically found in tropical lowlands around noon. We hypothesized that velamen radicum effectively insulates the living part from incident radiation. Therefore, we expected to find correlations between velamen thickness and thermal insulation traits, such as the maximal insulation strength (i.e., the temperature difference between the velamen surface and the outermost layer of the living cortex), and we tested this for 18 species of epiphytic Orchidaceae and Araceae. Additionally, as roots of epiphytic plants are frequently exposed to high radiation and any insulating effect is probably temporary, we hypothesized that the living tissues of aerial roots show considerable tolerance to high temperatures and that they can recover after heat exposure.

## 2. Results

### 2.1. Root and Velamen Anatomy

The roots of the studied Orchidaceae and Araceae were invariably velamentous, but there was considerable variation in thickness, both in absolute and relative terms. In absolute terms, the velamina of the studied Orchidaceae were, on average, three times thicker than those of Araceae (KW test, X^2^ = 6.7, df = 1, *p* < 0.01; Table 1). In relative terms, comparing the thickness of velamen and living tissue (i.e., parenchyma and central cylinder), the orchid velamen also accounted for a higher proportion of the total diameter. Values ranged from 39% for *Myrmecophila tibicinis* to 2% for *Anthurium hacumense* or *A. paludosum* (Table 1).

### 2.2. Effectiveness of the Velamen Radicum as Thermal Insulator

The time to reach the maximal insulation strength (i.e., insulation time, t_ins_) varied from 20–72 min (Table 2) in Orchidaceae and from 26–42 min in Araceae (KW test, X^2^ = 0.1, df = 1, *p* = 0.74). The mean surface temperature maxima (T_max_) were significantly lower for Orchidaceae compared to Araceae (41 vs. 49 °C; KW test, X^2^ = 0.1, df = 1, *p* < 0.001; Figure 1A), as was the maximal insulation strength (ΔT_max_; 1.6 vs. 2.8 °C; ANOVA, F = 35.2, df = 1, *p* < 0.001; Figure 1B). The somewhat shorter cooling period (t_cool_) in Orchidaceae was not significantly different from that in Araceae (23.2 vs. 19.2 min; KW test, X^2^ = 1.7, df = 1, *p* = 0.19; Figure 1C).

### 2.3. Influence of Velamen Thickness on Optical and Temperature Insulation Traits

Unexpectedly, ΔT_max_ decreased with velamen thickness (Spearman test, *p* < 0.01, R^2^ = 0.2; Figure 2A). However, a relevant characteristic of an insulating material is the delay in temperature change. In this respect, velamen thickness correlated positively with increasing t_ins_ (Spearman test, df = 14, *p* < 0.05, R^2^ = 0.7; Figure 2B) and t_cool_ (Spearman test, df = 14, *p* < 0.01, R^2^ = 0.3; Figure 2C). Further correlations among the insulation variables and the velamen thickness were negative, but weak (see Appendix A).

### 2.4. Heat Tolerance of Velamentous Roots

To test the temperature tolerance of roots, we subjected root sections to a series of heat treatments. The average root viability immediately after temperature exposure (i.e., the metabolic activity of living root tissue after the temperature treatment) decreased by 25% in the 42 °C treatment in all species. When roots were heated to 48 °C, their viability index was undistinguishable from that of boiled roots (Figure 3A). This instantaneous response differed among individual species. Using logistic regressions, we estimated a 50% loss of viability to be reached between 41 and 48 °C (Table 3). Changes in viability during the five-day recovery period were species-specific, but none of the species showed a significant recovery (ANCOVA, df = (1, 124), F = 0.002, *p* = 0.9). There were significant differences between heat treatments (ANCOVA, df = (1, 124), F = 97.5, *p* < 0.01; Figure 3B).

## 3. Discussion

We documented the thermal insulation traits of the velamen radicum of 18 epiphytic species of Orchidaceae and Araceae to assess velamen’s capacity to function as a thermal insulator. Temperature increases affect the basic metabolic functions of any tissue [15]. However, information regarding heat stress specifically on root tissue is very scarce. To our knowledge, available data are all related to terrestrial roots and focus on crop species [16]. Some studies address the indirect effects of light on root growth [17,18,19]. In contrast to terrestrial roots, roots of bark epiphytes usually grow exposed on a host tree, which probably led Pridgeon [5] to suggest that velamen radicum functions as thermal insulator. Our results hardly support this notion. Although we found a small insulating effect, i.e., ΔT_max_, to vary from 0.6 to 3.2 °C (shown in *Phalaenopsis pulchra* and *Anthurium obtusum*, respectively) with an average of 2.0 °C, the temperature of the outermost part of the living cortex (T_par_) still increased to almost 50 °C (*A. brownii*) in Araceae and up to 43 °C (*P. pulchra* and *P. schilleriana*, Table 2) in Orchidaceae under the conditions of our experiment. Therefore, the insulation function of velamen was limited. Surprisingly, ΔT_max_ even tended to decrease with velamen thickness (Figure 2A). A possible explanation comes from experiments with insect tissues. There, heating rates are inversely related to reflection [20], and possible differences in reflectance of velamen that are unrelated to velamen thickness may at least partly explain this unexpected finding.

The selected temperature range for the heat stress experiment covered the maximum temperatures measured in the insulation experiment to test if these temperatures could be detrimental to velamentous roots. The average T_max_ of the living cortex observed in the insulation experiment was c. 44 °C for the 16 studied species (Table 2), which is very close to or even exceeding the heat-tolerance limit of the living tissue of the tested species. However, we also observed a substantial delay in heating of the living cortex (Table 2); on average, it took 30 min to reach these high temperatures. We would need in situ measurements of root temperatures to assess how our laboratory measurements relate to field conditions. Unfortunately, there are no other data on the heat tolerance of epiphyte roots to compare our results with. We can, however, compare our data with those reported by Ingram et al. [21] for several crop dicot plants growing in pots. These values are actually higher than, ours ranging from 45.3 to 57.7 °C. Nevertheless, a direct comparison is difficult because these studies used different methods, i.e., electrolyte leakage, to quantify heat tolerance. Comparing the t_ins_ of velamen radicum between the species of the studied families, the studied orchids heated up more slowly than the aroids, but the latter needed less time to cool down. Because epiphytic Araceae species grow mostly in less exposed sites, thermal insulation of the roots might be less relevant in this family. In contrast, Orchidaceae species have a much wider range of growing sites and often grow at exposed sites [22,23,24].

With the appropriate caution, considering the limitations of our study, we may still suggest that heat tolerance of aerial roots could be one possible agent behind the typically observed vertical stratification of epiphytes [25,26]. Besides light adaptation of the photosynthetic apparatus [27], crassulacean acid metabolism [28,29] or other aspects of plant water relations [30,31], the heat tolerance of their roots could be yet another explanatory factor that should be explored in future studies.

## 4. Materials and Methods

The roots of twelve orchid and six aroid species were sampled from plants cultivated in the greenhouses of the Carl von Ossietzky University Oldenburg, Germany, and the Botanical Garden Berlin (Botanischer Garten und Botanisches Museum Berlin, Freie Universität Berlin, Berlin, Germany), along with a few commercial species (see Appendix B). Growing conditions can be described as moist tropical lowland conditions with additional light during winter. Species names follow the checklist in [32]. After collection, roots from Berlin were kept moist in ventilated plastic bags until processing at the University of Oldenburg. Both thermal insulation trait measurements and heat-tolerance tests were made within five days after collection. All tests were performed using samples from sections at least three centimeters behind the root apex.

### 4.1. Root Anatomy Measurements

Velamen and parenchyma thickness, the diameter of the central cylinder and the number of cell layers in velamen were studied in cross-sections under a microscope using 4× and 10× magnification (POLYVAR, Reichert-Jung, Austria).

### 4.2. Temperature Measurements

To determine the insulating properties of velamen, we measured the temperature on the root surface and in the parenchyma of the roots as close as possible to the exodermis, i.e., the outermost layer of the living cortex. Specifically, an infrared (IR) camera (testo 875-1, Testo SE & Co. KGaA, Lenzkirch, Germany) placed at 10 cm from the root measured the surface temperature, while the temperature of the living tissue was measured with a needle probe (thermocouple OMEGA HYP-1, copper-constantan, needle diameter 0.3 mm, OMEGA Engineering Inc., Deckenpfronn, Germany) inserted in the parenchyma just below the velamen (Figure 4).

Roots were exposed to a. NIR light source (Theratherm light bulb R95, 230 V, 100 W, OSRAM, Munich, Germany). The lamp was installed level to the test object at a distance of about 20 cm (Figure 5). A thermal power sensor (S302C, ThorLabs Inc., Newton, NJ, USA) was used to verify that the test objects were exposed to irradiance levels (4000 W m^−2^) comparable to solar irradiance levels around noon in the open in the tropics [33,34]. Room temperature was continuously monitored during the experiments using dataloggers (Hobo U23-001, Onset, Bourne, MA, USA). Values ranged from 22 °C to 25 °C and relative humidity was about 50%.

Each root sample was placed in a small pit in a polystyrene plate (Figure 5). The thermocouple signal was transferred to a computer and a data point was stored every second. An IR camera reading was taken every minute. Each test series started with three minutes at an ambient temperature, i.e., the IR lamp was switched off. Immediately after the third reading, the IR lamp was switched on, and the root was exposed to IR light until no further temperature increase was detected by the thermocouple. Then, the IR lamp was switched off, and the cooling was recorded as described above until the ambient temperature was reached again. All tests were conducted with air-dried velamen. Because there was a difference of c. 1 °C between the IR camera and the needle probe record at the start of each run, we fitted the data from the thermocouple of every sample to obtain an initial difference of zero between the two instruments.

In the experiment, we assessed temporal changes in root temperature to detect a possible insulating effect of velamen. In Figure 6, we define several terms. We recorded the temperature on the surface (T_surf_) and immediately underneath (T_par_) velamen, i.e., the temperature of the outermost part of the living cortex. The insulation strength (ΔT) is the difference between the temperature on the surface and the temperature in the living tissues of the root. During the experiment, ΔT reached three different stabilization points. Without the IR lamp, T_surf_ and T_par_ were almost identical (i.e., ΔT was equal or close to zero, ΔT_o_). Upon exposure to irradiance both T_surf_ and T_par_ increased while ΔT varied during the heating process but stabilized towards the end of heating (ΔT_1_). Insulation time (t_ins_) is the period from the start until ΔT_1_ is reached, and the highest value of ΔT in this period is the maximum insulation strength (ΔT_max_), typically at the time of a constant high temperature registered on the velamen surface (T_max_). Once the IR lamp was turned off, roots started to cool immediately and returned to a near-ambient temperature, typically with a small temperature difference between velamen surface and cortex (ΔT_2_). The time to reach this state was dubbed the cooling period (t_cool_).

### 4.3. Heat-Tolerance Test

Root viability was assessed by staining root samples with the redox solution 0.6% 2,3,5- Triphenyltetrazolium chloride (TTC), potassium phosphate buffer (0.1 M, pH = 7) and Tween 20 (0.05%), which yielded the detection of metabolic activity of living tissue. We largely followed the protocol of Ruf and Brunner [35] and measured the absorbance of Tryphenylformazan (TF, reduced form of TTC) at 520 nm. Briefly, 50 mg of cut root slices of around 2 mm were soaked in TTC solution for 24 h at 30 °C in a dark chamber in 2 mL Eppendorf tubes. Later, the TTC solution was extracted, and samples were ground in a swing mill (MM 200, Retsch, Haan, Germany) with steel balls of 2 mm diameter for three minutes at 30 Hz. We added 2 mL ethanol (90%) to the ground material and vortexed the samples for 10 s to extract the TF. This solution was centrifuged for five minutes at 10,000 rpm (Biofuge fresco, Thermo Fisher Scientific, Schwerte, Germany) to finally measure the absorption of the supernatant at 520 nm in a UV-VIS 1202 Shimadzu spectrophotometer (Thermo Fisher Scientific, Schwerte, Germany). The precipitates (root residuals) were dried for 24 h at 80 °C and their dry weight (DW) was recorded. This yielded the estimation of root tissue viability as the absorbance of TF at 520 nm per gram DW (i.e., A_520_ g _DW_^−1^).

We performed heat-tolerance tests with a subset of species (n = 8, Appendix B). For that, we cut three root segments per treatment per species about 9 cm in length. We discarded the first three centimeters from the tip and avoided, as much as possible, branching roots. The cut ends were sealed with petroleum jelly to avoid water loss from the cuttings. For each heat treatment, roots were put in plastic bags and vacuumed with a vacuum sealer (VC10, Caso Design, Arnsberg, Germany). The bags were immersed for 15 min in preheated water baths (LAUDA CS20, Messgeräte-Werk LAUDA, Lauda-Königshofen, Germany). We selected a range of temperatures (42, 48, 51 and 54 °C) roughly covering the range of maximum temperatures measured in the thermal insulation experiment. Immediately after temperature exposure in the water bath, root samples were sliced from the ends of the treated roots (first discarding the petroleum jelly covered end) to proceed with the TTC protocol and the ends of the roots were sealed again with petroleum jelly to test root recovery over the following five days. This period was chosen to avoid artefactual results related to fungi contamination after the fifth day of treatment. The treated roots were kept at room temperature on humid filter paper in Petri dishes to prevent tissue damage due to desiccation. To assess root recovery, we repeated the TTC test after one and five days. There were two controls; one set of roots was kept at room temperature (approximately 22 °C) and another one was boiled at 100 °C. The room temperature control showed a slight decrease in viability after five days of monitoring, and the zero control, i.e., dead tissue, showed consistently low values of c. 25 A_520_ g _DW_^−1^ over the five days.

### 4.4. Data Analysis

Family differences in velamen thickness were analyzed with using the Kruskal–Wallis rank sum test. We analyzed relationships between several morphological root traits and thermal insulation properties of the roots. Velamen thickness and the number of velamen cell layers were highly correlated (Pearson’s product moment test, *p* = 0.01, r = 92%); therefore, we used only velamen thickness in the analyses.

We analyzed the insulation capacity of velamen and possible correlations of the insulation parameters (T_max_, ΔT_max_, t_ins_ and t_cool_) with velamen thickness for each species. We also assessed whether these parameters differed between families using categorical tests (ANOVA and KW). Because we studied only a small number of species of two large families, any differences or lack thereof should be interpreted with caution.

For the analysis of heat tolerance of velamentous roots, we calculated the percentage viability in the monitored days. For that, we used Equation (1) as follows:Viability (%) = A_520[i]_ − A_520[100 °C]_/A_520[22 °C]_ − A_520[100 °C]_ × 100(1)
where A_520[i]_ refers to the absorbance at a certain heat treatment; A520_22 °C_ refers to the absorbance at room temperature; and A520_100 °C_ refers to the absorbance at 100 °C. To analyze the recovery of roots over time we used a logarithm of the variable “viability” as a response variable in ANCOVA, with ‘species’ as the independent variable, and the variables ‘temperature’ and ‘time after treatment’ as covariates. We also estimated the temperature when root tissues lose 50% of their viability in every species, applying logistic regressions on our results (LT_50_).

All statistical analyses were performed with R 3.4.1 [36]. If not stated otherwise, means are accompanied by standard deviations (SD).

## 5. Conclusions

We found limited support for the notion that velamina act as thermal insulators. For roots with thin velamen, the fast heating goes along with fast cooling, while thick velamina take longer to heat up but also longer to cool down. This could benefit velamentous roots of shade species when they are suddenly, and usually for a short time only, exposed to sun flecks. Our results regarding heat tolerance of velamentous roots were inconclusive. However, our study demonstrates substantial species-specific differences in heat tolerance, which might be part of the mechanistic basis of the frequently observed vertical stratification of epiphyte assemblages.

## Figures and Tables

**Figure 1 plants-12-01695-f001:**
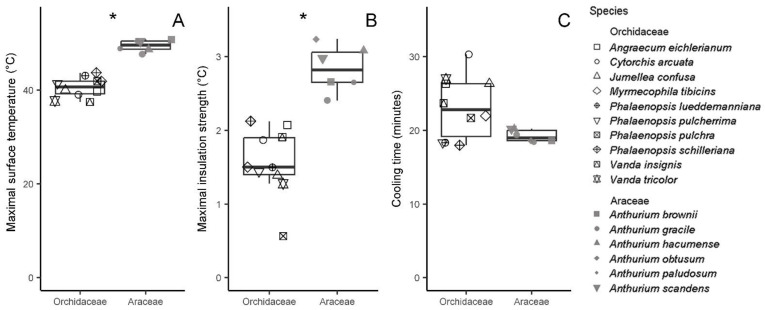
Comparison of thermal insulation traits: maximal surface temperature (T_max_, (**A**)), maximal insulation strength (ΔT_max_, (**B**)) and cooling time (t_cool_, (**C**)) between 16 species of Orchidaceae and Araceae. Significant differences are indicated with an asterisk (*p* < 0.05).

**Figure 2 plants-12-01695-f002:**
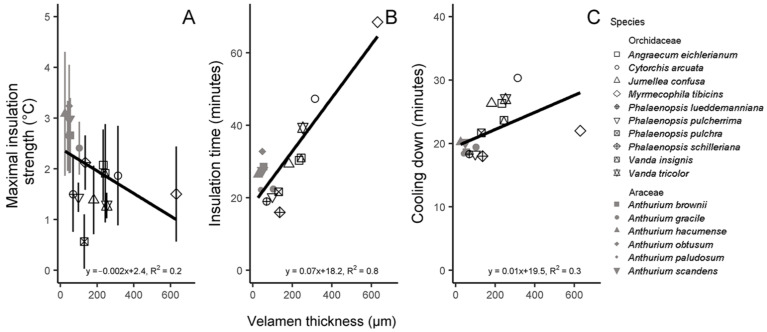
Thermal insulation traits correlated with the velamen thickness of 16 epiphyte species: the maximal insulation strength (ΔT_max_, (**A**)), the insulation time (**B**) and the time to cool down (**C**). Species belonged to Orchidaceae (black, open symbols) and Araceae (grey, closed symbols). The equations and coefficients of determination are given for each regression (all *p* < 0.05). Data are means of three individuals per species.

**Figure 3 plants-12-01695-f003:**
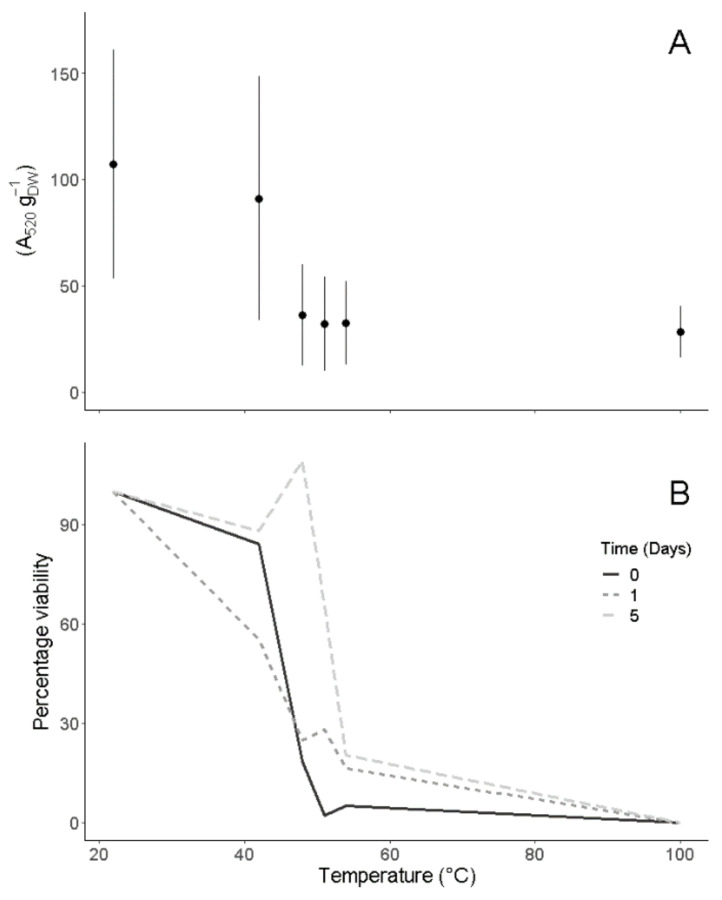
Viability of velamentous roots of eight species following heat treatments at 22, 42, 48, 51, 54 and 100 °C (**A**). Measurements were obtained immediately after every heat treatment. For species names, see Appendix B. The absorbance of Triphenylformazan (TF, reduced form of TTC) per gram of dry weight (DW) was measured at 520 nm (expressed as A_520_ g^−1^ _DW_). The 22 °C and 100 °C treatments were implemented as controls. Root tissue recovery was analyzed over five days (**B**). Data are means of three individuals per heat treatment.

**Figure 4 plants-12-01695-f004:**
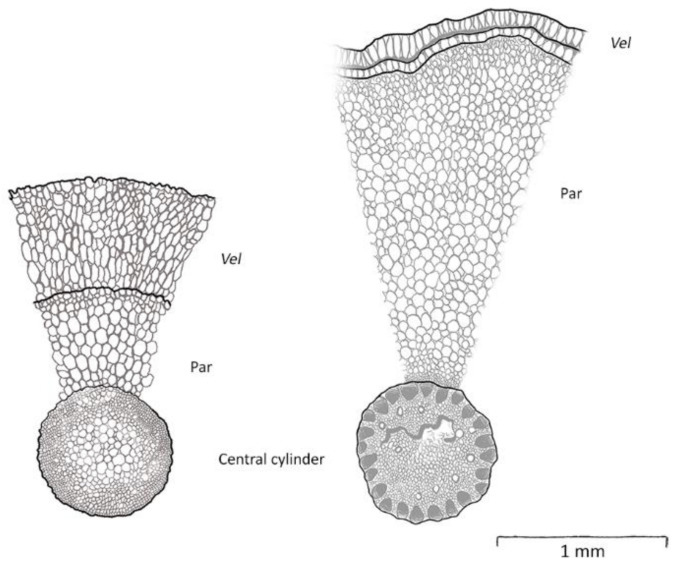
Schematic view of two crosscuts of velamentous roots, *Myrmecophila tibicins* (Orchidaceae) on the left and *Anthurium paludosum* (Araceae) on the right. Central cylinder, parenchyma (Par), and *velamen radicum* (*Vel*) are shown.

**Figure 5 plants-12-01695-f005:**
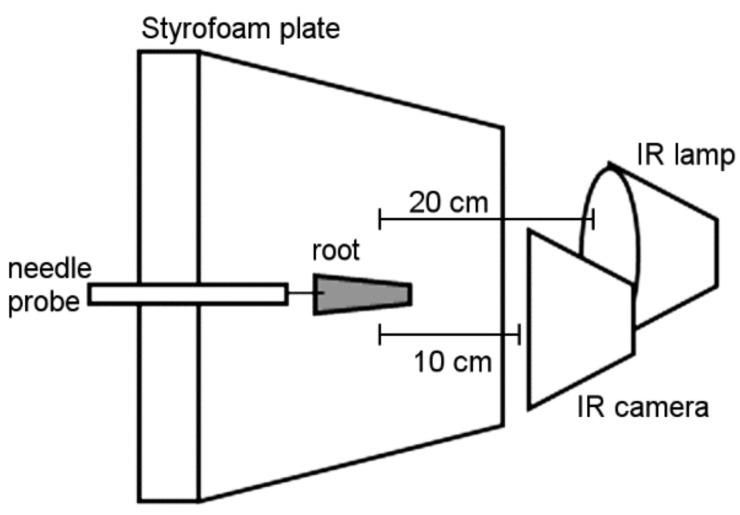
Experimental set-up of the temperature measurements. The infrared (IR) lamp was installed level with the test object at a 20 cm distance. The IR camera was installed level with the test object at a 10 cm distance.

**Figure 6 plants-12-01695-f006:**
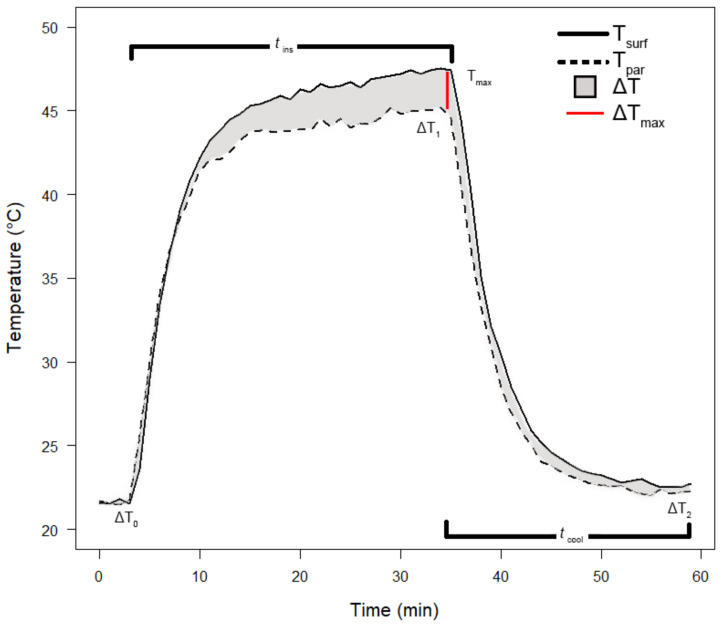
Representative run of the thermal insulation test. We defined insulation strength (ΔT) as the difference between the temperatures of the velamen surface (T_surf_) and the outermost layer of the living tissue (T_par_), the maximal difference being ΔT_max_. We determined the period of heating (t_ins_, infrared light switched on until ΔT_1_ was reached) and the period of cooling (t_cool_, infrared light switched off until ΔT_2_ was reached). At the start of t_ins_, the temperature difference was minimal (ΔT_o_), and typically maximal (ΔT_1_) at the end of t_ins_ with a stable maximal temperature (T_max_). The test ended when the curve flattened again after switching off the light source. At that point, the temperature difference (ΔT_2_) was again very small but usually larger than at the beginning (ΔT_o_).

**Table 1 plants-12-01695-t001:** Velamen thickness, number of velamen cell layers and the ratio (as percentage) of velamen to total root diameter (V ratio) of aerial roots of tested species. Data are means ± SD.

Family/Species	n		Velamen	
		Thickness (µm)	# Cell Layers	V Ratio (%)
Orchidaceae				
*Angraecum eichlerianum*	3	234 ± 23	5	7 ± 1
*Cyrtorchis arcuata*	3	314 ± 63	5 ± 1	10 ± 3
*Jumellea confusa*	3	182 ± 12	4	7 ± 1
*Myrmecophila tibicinis*	3	630 ± 16	8	39 ± 1
*Phalaenopsis lueddemanniana*	4	69 ± 30	2	4 ± 1
*Phalaenopsis pulcherrima*	3	99 ± 25	4	5 ± 1
*Phalaenopsis pulchra*	3	130 ± 28	3	7 ± 2
*Phalaenopsis schilleriana*	3	136 ± 71	2 ± 1	6 ± 2
*Vanda insignis*	3	244 ± 23	5 ± 1	10 ± 1
*Vanda tricolor*	3	252 ± 30	5 ± 1	10 ± 1
Araceae				
*Anthurium brownii*	3	54 ± 7	2	5
*Anthurium gracile*	3	103 ± 23	4 ± 1	9 ± 2
*Anthurium hacumense*	3	25 ± 5	1	2 ± 1
*Anthurium obtusum*	3	48 ± 11	1	3 ± 1
*Anthurium paludosum*	3	41 ± 7	1 ± 1	2
*Anthurium scandens*	3	49 ± 6	1	5 ± 1

**Table 2 plants-12-01695-t002:** Parameters determined in the thermal insulation test (for definition of parameters, see materials and methods). The means of the maximal temperature (T_max_) on the root surface (T_surf_) and in the outermost part of the parenchyma (T_par_), the maximal insulation strength (ΔT_max_), as well as the time it took roots to reach ΔT_max_ (t_ins_) and the time it took them to cool down (t_cool_), are shown. Data are means ± SD among replicates.

Family/Species	n	T_max_ (°C)		ΔT_max_ (°C)	t_ins_ (min)	t_cool_ (min)
		T_surf_	T_par_			
Orchidaceae						
*Angraecum eichlerianum*	3	39.7	38.3	2.1	30.3 ±0.7	26.3 ± 0.7
*Cyrtorchis arcuata*	3	39	37.9	1.9	47.3 ± 1	30.3 ± 0.5
*Phalaenopsis pulcherrima*	3	41.5	41.3	1.4	20.3 ± 0.3	18.3 ± 0.3
*Jumellea confusa*	3	40	39.3	1.4	29.3 ±0.7	26.3 ± 0.5
*Phalaenopsis lueddemanniana*	3	43.1	42.4	1.5	19 ± 0.7	18.3 ± 1
*Phalaenopsis pulchra*	3	41.9	42.1	0.6	21.7 ± 0.5	21.7 ± 0.1
*Phalaenopsis schilleriana*	3	43.7	43.3	2.1	16 ± 0.5	18 ± 1.4
*Myrmecophila tibicinis*	3	41.9	43.3	1.5	68.5 ±0.9	22 ± 0.1
*Vanda insignis*	3	37.5	36.9	1.9	31 ± 1	23.7 ± 0.3
*Vanda tricolor*	3	37.7	37.9	1.3	39.3 ±0.3	27 ± 0.6
Araceae						
*Anthurium brownii*	5	50.8	49.1	2.7	28.6 ± 0.7	18.6 ± 1
*Anthurium gracile*	5	47.7	46.1	2.4	22.4 ± 0.5	19.4 ± 1
*Anthurium hacumense*	5	48.7	47.1	3.1	26.4 ± 1.2	20.2 ± 0.8
*Anthurium obtusum*	5	50.5	48.3	3.2	32.8 ± 0.7	18.6 ± 0.8
*Anthurium paludosum*	5	48.9	47.4	2.6	22.2 ± 0.7	18.4 ± 0.8
*Anthurium scandens*	5	50.4	48.8	3.0	27.6 ± 1.1	20.2 ± 1.1
Total average		43.9	43.1	2.0	28.8	21.2

**Table 3 plants-12-01695-t003:** Estimated temperature when roots lose 50% viability (LT_50_). Results were obtained from viability measurements on roots tested immediately after every heat treatment. Temperature estimates were calculated with logarithmic equations describing the dependence of viability on the temperature of the heat treatments (from 22 to 100 °C).

Species	Temperature (°C)
	LT_50_
*Anthurium brownii*	46
*Anthurium gracile*	43
*Anthurium hacumense*	45
*Anthurium obtusum*	41
*Anthurium paludosum*	44
*Anthurium scandens*	44
*Epidendrum nocturnum*	44
*Phalaenopsis* sp.	48

## Data Availability

Data will be made available upon request.

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
