# Peer review of "Does A Velamen Radicum Effectively Protect Epiphyte Roots against Excessive Infrared Radiation?"

_plants, 2023, doi:10.3390/plants12081695_

Round 1

Reviewer 1 Report

The communication study ’Does a velamen effectively protect epiphyte roots against excessive radiation? (no. 2332250)’ tells the the potential role of velamen tissue found in epiphytic plants. Is it providing protection against excessive infrared radiation in the upper forest canopy? The study investigated the insulation properties of velamina in 18 Orchidaceae and Araceae species and found limited support for its insulating function. The results show that while there is a small insulating effect, the insulation function of velamen is limited. The study suggests that heat tolerance of aerial roots could be a possible agent behind the typically observed vertical distribution of epiphytes in the canopy. The study is basically well-written. However, authors (starting from the title) use the phrase ’excess radiation’. This is confusing for the readers, because they performed heat stress on orchid plants. Radiation covers wide range of electromagnetic field. Authors must to clarify and modify it to either ’excessive infrared radiation’ or simply ’heat stress’. The Result description and illustration of data are adequate. The number of figures is plenty, they properly provide data. There are some message in the text ’Error! Reference source not found’ – they must be corrigated before publishing. From the beginning of page 12, ’ ).’ must be removed. On the figure of Appendix A, correlation plot, indicating non-significant fields using ’X’ is not correct. Authors should re-edit this plot, and using  asterisks for significance levels instead of ’X’. The R code can do that. Regarding, that this is a short study without any deeper biological analyses, it can be considered for publication in Plants.

Author Response

Reply to the review report

The communication study ’Does a velamen effectively protect epiphyte roots against excessive radiation? (no. 2332250)’ tells the potential role of velamen tissue found in epiphytic plants. Is it providing protection against excessive infrared radiation in the upper forest canopy?

With this first assessment of the velamen properties as a thermal insulator, we found a limited function of velamen as such an insulator. We propose that further studies on additional epiphytic species with velamen will confirm that the velamen indeed protects the living parts of roots from excessive infrared radiation in the upper forest canopy.

The study investigated the insulation properties of velamina in 18 Orchidaceae and Araceae species and found limited support for its insulating function. The results show that while there is a small insulating effect, the insulation function of velamen is limited. The study suggests that heat tolerance of aerial roots could be a possible agent behind the typically observed vertical distribution of epiphytes in the canopy. The study is basically well-written. However, authors (starting from the title) use the phrase ’excess radiation’. This is confusing for the readers, because they performed heat stress on orchid plants. Radiation covers wide range of electromagnetic field. Authors must to clarify and modify it to either ’excessive infrared radiation’ or simply ’heat stress’.

The approach of our study was two-fold. The first aspect tested the insulation effect of velamen when illuminated with IR, while the second part asks the question whether the root tissue recovers after heat stress. We agree to be more specific and rephrase our title to “…excessive infrared radiation.”

The Result description and illustration of data are adequate. The number of figures is plenty, they properly provide data. There are some message in the text ’Error! Reference source not found’ – they must be corrigated before publishing.

We checked that cross-references works adequately.

From the beginning of page 12, ’ ).’ must be removed. On the figure of Appendix A, correlation plot, indicating non-significant fields using ’X’ is not correct. Authors should re-edit this plot, and using  asterisks for significance levels instead of ’X’. The R code can do that.

We solved this issue by rearranging the placement of Appendix A. Regarding the use of the “X” symbol on Appendix A, we kept this symbol on the plot instead of an asterisk for results lacking of significance. Since results with significant relationship are greater than those with no significance, the plot would be oversaturated with asterisks and therefore it would not look well designed.

Regarding, that this is a short study without any deeper biological analyses, it can be considered for publication in Plants.

We appreciate your comments.

Reviewer 2 Report

1.       The current study investigated the possible insulating effect of the velamen by comparing the temperature increases of the velamen surface and the outermost layer of the living cortex when exposed to near-infrared radiation (NIR) of an intensity typically found in tropical lowlands around noon. It is a very interesting scientific question. Why the authors chose two shade plants, Orchidaceae and Araceae, as your study materials? Why not choose a heliophyte and a shade plant for controlled studies will be more different.

2.       All tables or figures are placed on the same page;

3.       Some minor problems I have marked in yellow in the manuscript, please correct.

1.       The current study investigated the possible insulating effect of the velamen by comparing the temperature increases of the velamen surface and the outermost layer of the living cortex when exposed to near-infrared radiation (NIR) of an intensity typically found in tropical lowlands around noon. It is a very interesting scientific question. Why the authors chose two shade plants, Orchidaceae and Araceae, as your study materials? Why not choose a heliophyte and a shade plant for controlled studies will be more different.

2.       All tables or figures are placed on the same page;

3.       Some minor problems I have marked in yellow in the manuscript, please correct.

1.       The current study investigated the possible insulating effect of the velamen by comparing the temperature increases of the velamen surface and the outermost layer of the living cortex when exposed to near-infrared radiation (NIR) of an intensity typically found in tropical lowlands around noon. It is a very interesting scientific question. Why the authors chose two shade plants, Orchidaceae and Araceae, as your study materials? Why not choose a heliophyte and a shade plant for controlled studies will be more different.

2.       All tables or figures are placed on the same page;

3.       Some minor problems I have marked in yellow in the manuscript, please correct.

1.       The current study investigated the possible insulating effect of the velamen by comparing the temperature increases of the velamen surface and the outermost layer of the living cortex when exposed to near-infrared radiation (NIR) of an intensity typically found in tropical lowlands around noon. It is a very interesting scientific question. Why the authors chose two shade plants, Orchidaceae and Araceae, as your study materials? Why not choose a heliophyte and a shade plant for controlled studies will be more different.

2.       All tables or figures are placed on the same page;

3.       Some minor problems I have marked in yellow in the manuscript, please correct.

1.       The current study investigated the possible insulating effect of the velamen by comparing the temperature increases of the velamen surface and the outermost layer of the living cortex when exposed to near-infrared radiation (NIR) of an intensity typically found in tropical lowlands around noon. It is a very interesting scientific question. Why the authors chose two shade plants, Orchidaceae and Araceae, as your study materials? Why not choose a heliophyte and a shade plant for controlled studies will be more different.

2.       All tables or figures are placed on the same page;

3.       Some minor problems I have marked in yellow in the manuscript, please correct.

Author Response

Reply to the review report

Comments and Suggestions for Authors

  1. The current study investigated the possible insulating effect of the velamen by comparing the temperature increases of the velamen surface and the outermost layer of the living cortex when exposed to near-infrared radiation (NIR) of an intensity typically found in tropical lowlands around noon. It is a very interesting scientific question. Why the authors chose two shade plants, Orchidaceae and Araceae, as your study materials? Why not choose a heliophyte and a shade plant for controlled studies will be more different.

This study represents a first assessment of the velamen as a thermal insulator. For that, we used the plant material available considering feasibility, source, and time. We agree with the reviewer that there is certainly room for future studies.

  1. All tables or figures are placed on the same page;

We edited our manuscript accordingly.

  1. Some minor problems I have marked in yellow in the manuscript, please correct.

Solved. We appreciate your suggestions.

Reviewer 3 Report

·         I have doubts about the question format of the title.

·         The author should use past tense for things you did, and present for general theories and applications.

·         Material and methods, it would be interesting to present information on plant cultivation before harvesting (irrigation, nutrient, and climate properties).

·         What is (Error! Reference source not found.)?

Author Response

Reply to the Review Report

I have doubts about the question format of the title.

We changed our title to “Does a velamen effectively protect epiphyte roots against excessive infrared radiation?”

The author should use past tense for things you did, and present for general theories and applications.

We made the pertinent changes.

Material and methods, it would be interesting to present information on plant cultivation before harvesting (irrigation, nutrient, and climate properties).

We do not have detailed information on growth conditions, but we at least give a short statement that the plants were kept under tropical lowland conditions with some additional light during winter when days are short.

  • What is (Error! Reference source not found.)?

We corrected this.